# Structural basis for potent and broad inhibition of HIV-1 RT by thiophene[3,2-d]pyrimidine non-nucleoside inhibitors

Yang Yang[1,2†], Dongwei Kang[3†], Laura A Nguyen[1], Zachary B Smithline[1], Christophe Pannecouque[4], Peng Zhan[3*], Xinyong Liu[3*], Thomas A Steitz[1,2,5*]

[1]Department of Molecular Biophysics and Biochemistry, Yale University, New Haven, United States; [2]Howard Hughes Medical Institute, Yale University, New Haven, United States; [3]Department of Medicinal Chemistry, Key Laboratory of Chemical Biology (Ministry of Education), School of Pharmaceutical Sciences, Shandong University, Jinan, China; [4]Rega Institute for Medical Research, KU Leuven, Leuven, Belgium; [5]Department of Chemistry, Yale University, New Haven, United States

**Abstract** Rapid generation of drug-resistant mutations in HIV-1 reverse transcriptase (RT), a prime target for anti-HIV therapy, poses a major impediment to effective anti-HIV treatment. Our previous efforts have led to the development of two novel non-nucleoside reverse transcriptase inhibitors (NNRTIs) with piperidine-substituted thiophene[3,2-d]pyrimidine scaffolds, compounds K-5a2 and 25a, which demonstrate highly potent anti-HIV-1 activities and improved resistance profiles compared with etravirine and rilpivirine, respectively. Here, we have determined the crystal structures of HIV-1 wild-type (WT) RT and seven RT variants bearing prevalent drug-resistant mutations in complex with K-5a2 or 25a at ~2 Å resolution. These high-resolution structures illustrate the molecular details of the extensive hydrophobic interactions and the network of main chain hydrogen bonds formed between the NNRTIs and the RT inhibitor-binding pocket, and provide valuable insights into the favorable structural features that can be employed for designing NNRTIs that are broadly active against drug-resistant HIV-1 variants.
DOI: https://doi.org/10.7554/eLife.36340.001

**\*For correspondence:**
zhanpeng1982@sdu.edu.cn (PZ);
xinyongl@sdu.edu.cn (XL);
thomas.steitz@yale.edu (TAS)

†These authors contributed equally to this work

**Competing interests:** The authors declare that no competing interests exist.

## Introduction

HIV-1 reverse transcriptase (RT) (hereinafter referred to as RT) plays an essential role in the viral life cycle by reverse transcribing the single-stranded RNA genome to a double-stranded DNA copy (*Deeks et al., 2015*; *Engelman and Cherepanov, 2012*). For this reason, it has been an important target of anti-HIV therapies (*Esté and Cihlar, 2010*; *Gubernick et al., 2016*). There are two main types of RT inhibitors: nucleoside RT inhibitors (NRTIs), which act as chain terminators and compete with incoming nucleotides in the polymerase active site (*Ren et al., 1998*; *Sarafianos et al., 1999*; *Tu et al., 2010*; *Yarchoan et al., 1988*), and non-nucleoside RT inhibitors (NNRTIs), which inhibit the activity of RT noncompetitively (*Merluzzi et al., 1990*; *Spence et al., 1995*). NNRTIs are a group of structurally diverse compounds that bind to the non-nucleoside inhibitor-binding pocket (NNIBP) located ~10 Å from the polymerase active site (*Ding et al., 1995*; *Kohlstaedt et al., 1992*; *Ren et al., 1995*). NNIBP is a hydrophobic pocket that emerges only when NNRTIs bind and induce conformational rearrangements of the residues defining the pocket (*Esnouf et al., 1995*; *Hsiou et al., 1996*). NNRTIs are key components in highly active antiretroviral therapy (HAART) due to their high specificity, desirable pharmacokinetics and generally good tolerance (*Moore and Chaisson, 1999*; *Pomerantz and Horn, 2003*).

Despite the success of NNRTIs in suppressing HIV-1 replication and reducing viral loads, their effectiveness is compromised by the emergence of drug-resistant mutations in RT (*Wainberg et al., 2011*). Earlier NNRTIs, including nevirapine (NVP), delavirdine (DLV) and efavirenz (EFV), have low genetic barriers for resistance and are extremely susceptible to mutations in the NNIBP of RT (*Das and Arnold, 2013a*, *2013b*). K103N, Y181C and Y188L are among the most prevalent NNRTI-resistant mutations identified in RT (*de Béthune, 2010*; *Wensing et al., 2017*). Y181C and Y188L mutations introduce steric hindrances between NNRTIs and the pocket, and/or eliminate critical π-π stacking interactions between side chains of the two tyrosine residues and the aromatic rings in NNRTIs (*Hsiou et al., 1998*; *Ren et al., 2001*). As to the K103N mutation, it was long believed that it prevented the entry of NNRTIs by stabilizing the closed conformation of NNIBP (*Hsiou et al., 2001*). However, a more recent study indicates that the resistance is more likely caused by the electrostatic difference between Asn103 and Lys103 (*Lai et al., 2016*). In the light of this new piece of data, the K103N mutation seems to utilize the same mechanism as Y181C and Y188L do to confer resistance to NNRTIs: by altering the shape or surface property of the NNIBP.

Next-generation NNRTIs are designed with conformational flexibility and positional adaptability and are able to target the NNIBPs of an array of drug-resistant RT mutants (*Das et al., 2008*, *2004*). Etravirine (ETR, also known as TMC125) and rilpivirine (RPV, also known as TMC278) are two U.S. Food and Drug Administration (FDA)-approved second-generation NNRTIs belonging to the diarylpyrimidine (DAPY) family (*Figure 1*). Both drugs show potent antiviral activities against wild-type (WT) HIV-1 and many HIV-1 variants displaying significant resistance to first-generation NNRTIs (*Janssen et al., 2005*; *Ludovici et al., 2001*). However, some existing resistance-associated RT mutations, such as K101P and Y181I, can still cause substantial decreases in susceptibility to ETR and RPV (*Azijn et al., 2010*; *Giacobbi and Sluis-Cremer, 2017*; *Smith et al., 2016*). Besides, new resistant mutations can arise from prolonged use of ETR and RPV, which undermine their anti-HIV-1 activities (*Wensing et al., 2017*). In patients who failed ETR- or RPV-based therapies, E138K/Q/R are among the most frequently occurred mutations identified in RT (*Xu et al., 2013*). Therefore, it is imperative to develop new NNRTIs with improved drug-resistance profiles.

Our previous efforts have led to the design and synthesis of two piperidine-substituted thiophene [3,2-*d*]pyrimidine NNRTIs using ETR as a lead compound (*Figure 1*) (*Kang et al., 2017*, *2016*). Compound K-5a2 features a thiophene[3,2-*d*]pyrimidine central ring, and replaces the cyanophenyl right wing of ETR with a more extended piperidine-linked benzenesulfonamide group, while keeping the 4-cyano-2,6-dimethylphenyl structure in the left wing of ETR. Compound 25a shares the same central ring and right wing structures with K-5a2, but grafts the 4-cyanovinyl-2,6-dimethylphenyl structure of RPV onto its left wing. Compared with ETR, compound K-5a2 displays much lower cytotoxicity and increased anti-HIV-1 potency against WT virus and virus strains with a variety of NNRTI-resistant mutations, except K103N and K103N/Y181C (*Kang et al., 2016*). The further optimized compound 25a is exceptionally potent in inhibiting WT HIV-1 and exhibits significantly better anti-HIV-1 activities than ETR against all of the tested NNRTI-resistant HIV-1 strains in cellular assays (*Kang et al., 2017*).

In this study, we demonstrated that 25a is superior to RPV in inhibiting RT bearing a wide range of resistance mutations, including K101P, Y181I and K103N/Y181I, against which RPV loses considerable potency, and determined the crystal structures of WT and mutant RTs in complex with either K-5a2 or 25a. These structures illustrate the detailed interactions between RT and the two inhibitors, and explain why K-5a2 and 25a are resilient to NNRTI-resistant mutations in the NNIBP. Additionally, comparison of the binding modes of K-5a2 and 25a with those of ETR and RPV suggests the possible mechanisms for the susceptibilities of ETR and RPV to E138K and K101P mutations. Our results outline the structural features of NNRTIs that can be employed for future drug design to overcome prevalent NNRTI-resistant mutations.

## Results

### Structure determination

The complexes of 25a or K-5a2 bound to WT RT or RT variants with drug-resistant mutations were prepared by soaking either NNRTI into the RT crystals. The structures were determined by molecular replacement using the structure of WT RT/RPV complex (PDB ID: 4G1Q) as the search template and

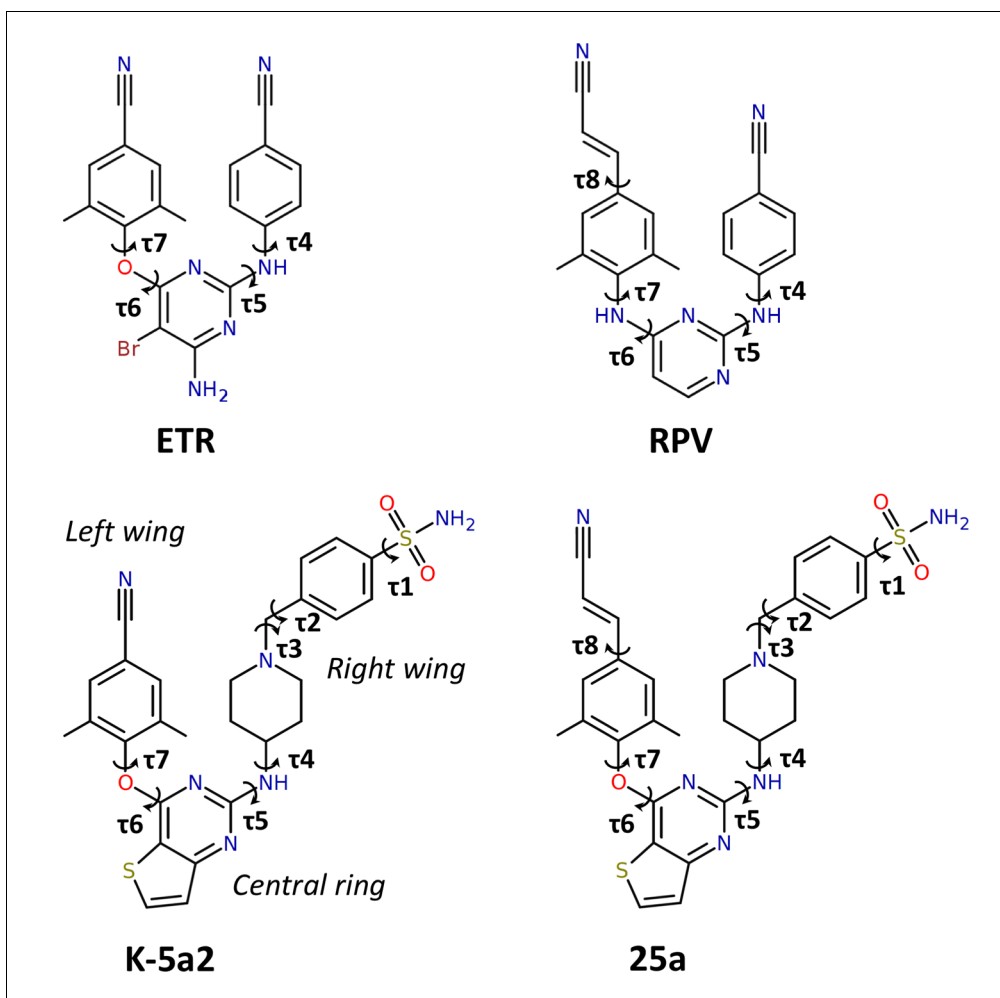

**Figure 1.** Chemical structures of NNRTIs. The torsion angles defining the rotatable bonds are labeled as τ1 to τ7 in K-5a2 and τ1 to τ8 in 25a. The equivalent torsion angles in ETR and RPV are labeled as τ4 to τ7 and τ4 to τ8, respectively. The structures of K-5a2 and 25a can be divided into three functional regions: a thiophene[3,2-*d*] pyrimidine central ring, a piperidine-linked benzenesulfonamide right wing, and a 4-cyano- (or 4-cyanovinyl-) 2,6-dimethylpheyl left wing.

DOI: https://doi.org/10.7554/eLife.36340.002

were subsequently refined to 1.9–2.23 Å resolution (*Supplementary file 1*). Overall, the structure of RT in the complexes has the same 'open-cleft' conformation as observed in prior RT/NNRTIs structures (*Figure 2A and C*) (*Das et al., 2008*, *2004*; *Ding et al., 1995*; *Ren et al., 1995*). The $F_o$-$F_c$ electron-density maps unambiguously defined the binding positions and conformations of both inhibitors in the NNIBP (*Figure 2B and D* and *Figure 2—figure supplement 1*).

## Interactions between piperidine-substituted thiophene[3,2-*d*]pyrimidine NNRTIs and RT

The RT-bound 25a and K-5a2 adopt a horseshoe conformation, which is similar to that seen with NNRTIs in the DAPY family (*Das et al., 2008*, *2004*). Both inhibitors exhibit remarkable structural complementarity to the NNIBP with substantial extensions into the three channels (tunnel, entrance and groove) characterizing the pocket (*Figure 3A and B* and *Figure 3—figure supplement 1A and B*). The left wing structures of 25a and K-5a2 form hydrophobic interactions with Pro95 and Leu234, and project into the tunnel lined by Tyr181, Tyr188, Phe227, and Trp229, forming π-π interactions with these residues. The entrance channel gated by Glu138 in the p51 subunit and Lys101 in the p66 subunit is an underexplored region in the NNIBP. By substituting the central pyrimidine ring of

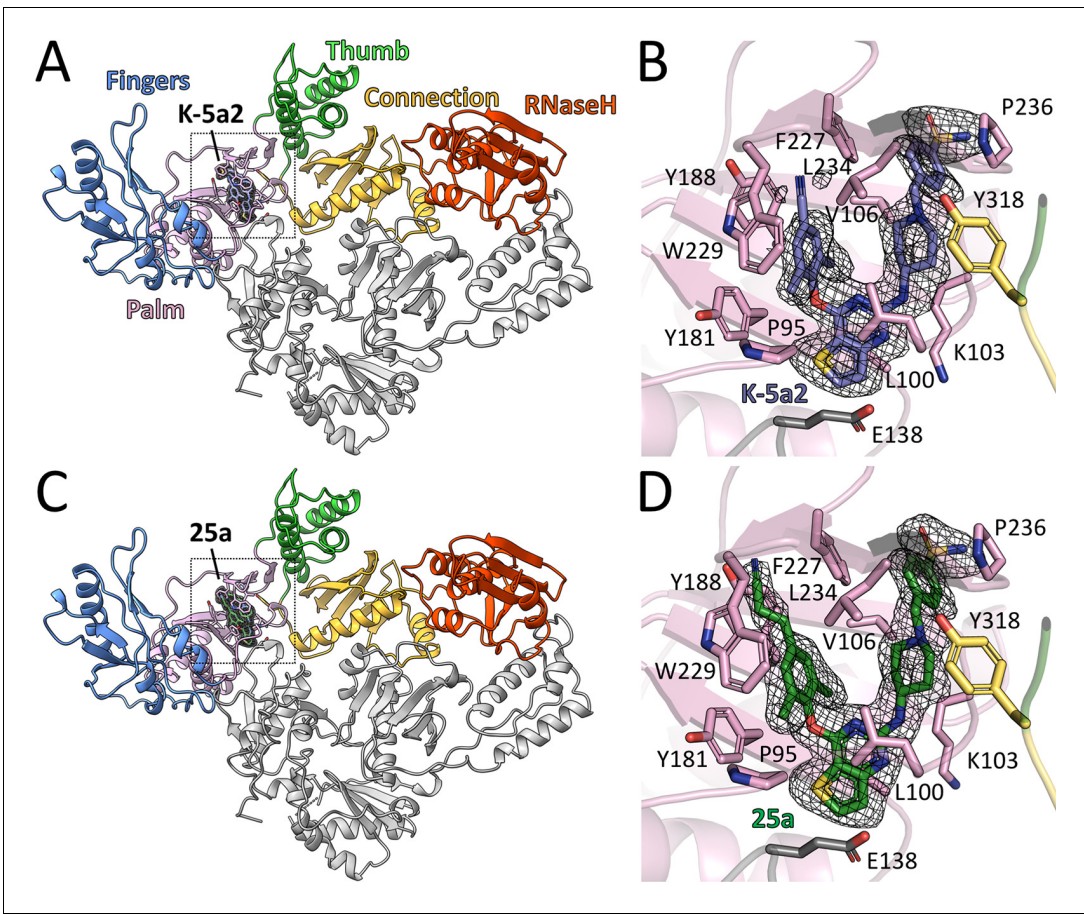

**Figure 2.** Structure of HIV-1 RT in complex with compound K-5a2 and 25a. (**A**) and (**C**) Overall structure of the HIV-1 WT RT in complex with compound K-5a2 determined at 1.92 Å resolution (**A**) and with compound 25a determined at 2.0 Å resolution (**C**). The p51 subunit is colored in gray, the fingers domain of the p66 subunit is colored in light blue, palm domain in pink, thumb domain in light green, connection domain in yellow, RNase H domain in red. Compound K-5a2 is in dark blue and compound 25a is in dark green. (**B**) and (**D**) An enlarged view of compound K-5a2 (**B**) and compound 25a (**D**) in the NNIBP with contacting residues shown as sticks. Compound K-5a2 and 25a are superposed with the electron density of their respective $F_o$–$F_c$ omit map (sharpened by applying a *B*-factor correction of –35 and contoured at 2.7σ).

DOI: https://doi.org/10.7554/eLife.36340.003

The following figure supplement is available for figure 2:

**Figure supplement 1.** Structures of different RT mutants in complex with K-5a2 or 25a.

DOI: https://doi.org/10.7554/eLife.36340.004

DAPY NNRTIs to a thiophene[3,2-*d*]pyrimidine heterocyclic structure, 25a and K-5a2 are able to establish nonpolar interactions with the alkyl chain of Glu138, while retaining the favorable hydrophobic contacts with Val179 and Leu100 manifested in the complexes of RT with ETR or RPV (*Das et al., 2008*, *2004*). The piperidine-linked aryl structure of the right wing arches into the groove surrounded by Lys103, Val106, Pro225, Phe227, Pro236, and Tyr318, developing numerous van der Waals contacts with their lipophilic side chains, and directs the terminal sulfonamide group to the solvent-exposed surface of RT.

In addition, binding of 25a and K-5a2 to the NNIBP is stabilized by an extensive hydrogen-bonding network between the inhibitors and the main chains of several key residues around the pocket (*Figure 3C and D* and *Figure 3—figure supplement 1C and D*): (i) the surface-positioned sulfonamide group is double-hydrogen bonded with the carbonyl oxygen of Lys104 and the backbone nitrogen of Val106; (ii) the piperidine nitrogen forms hydrogen bonds with the main chains of Lys103

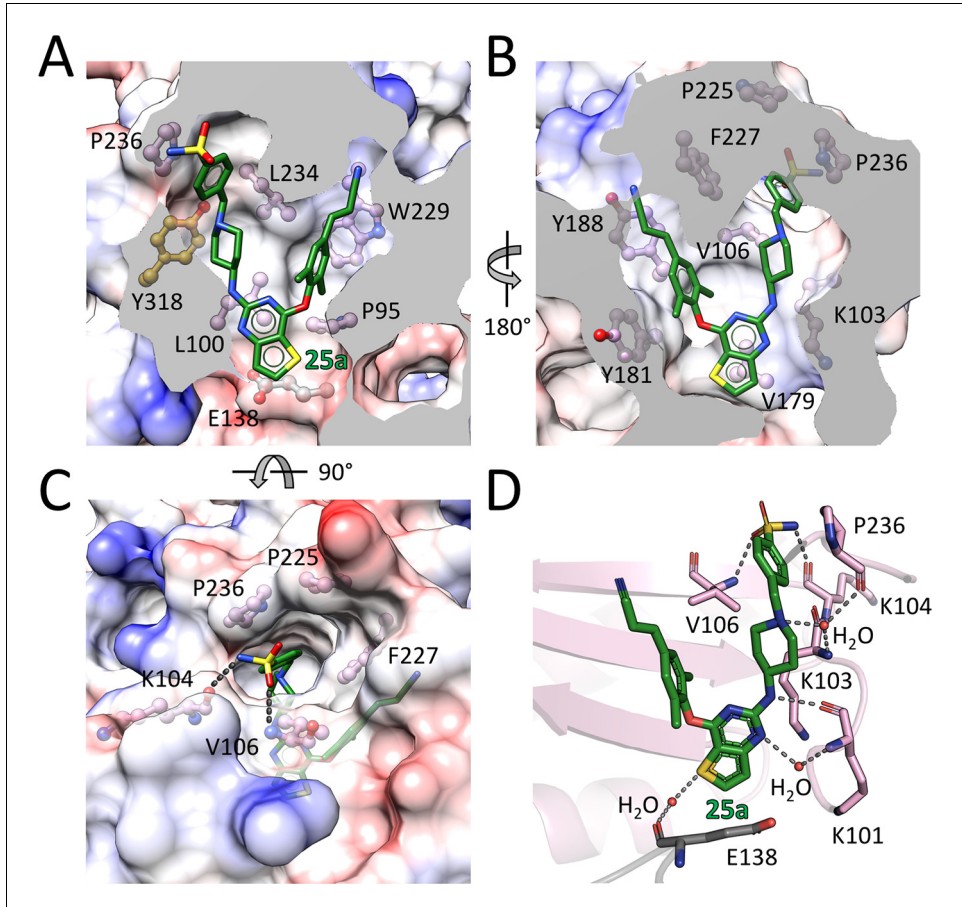

**Figure 3.** Detailed interactions between WT RT and compound 25a. (A) Back, (B) front and (C) top views of compound 25a in the NNIBP of RT. The inhibitor-binding pocket is in surface presentation and key residues are depicted as sticks. (D) Hydrogen bonds between compound 25a and the main chains of NNIBP residues. Residues in the p51 subunit are colored in gray, and residues in the p66 subunit are colored in pink.

DOI: https://doi.org/10.7554/eLife.36340.005

The following figure supplement is available for figure 3:

**Figure supplement 1.** Detailed interactions between WT RT and compound K-5a2.

DOI: https://doi.org/10.7554/eLife.36340.006

and Pro236 through a bridging water molecule; (iii) the amine group linking the central thiophene pyrimidine and the piperidine ring interacts with the carbonyl oxygen of Lys101, forming a conserved hydrogen bond observed in a number of second-generation NNRTIs/RT complexes (*Das et al., 2008*; *Lansdon et al., 2010*); (iv) additionally, the nitrogen and sulfur atoms in the central thiophene pyrimidine ring are involved in two water-mediated hydrogen bonds with the backbone nitrogen of Lys101 and the carbonyl oxygens of Glu138, respectively. These extensive interactions between the two piperidine-substituted thiophene[3,2-*d*]pyrimidine NNRTIs and RT lock the enzyme in an open-cleft conformation and inhibit its polymerization activity.

The above interactions between RT and the two NNRTIs generally agree with the results from molecular docking (*Kang et al., 2017*, *2016*). Nevertheless, a close inspection of the inhibitors observed in the crystal structures and those predicted by molecular docking reveals a few notable differences in their binding modes. First, the thiophene pyrimidine nitrogen in the inhibitors is not directly hydrogen bonded with the backbone nitrogen of Lys101, as predicted by inhibitor docking, but through a bridging water molecule instead. Second, the crystal structures define a water-mediated hydrogen bond between the carboxyl oxygen of Glu138 and the sulfur group in the inhibitors central ring, which is absent in the predicted binding modes. Due to their free movement and transient involvement in the binding process, it is difficult to predict the role of solvent molecules in the

interactions between enzymes and inhibitors using ligand-docking programs. These water-mediated interactions, however, can be critical for the enzyme–inhibitor complex formation and thus can provide important insights in understanding the resistance mechanisms of RT mutants.

## Inhibition of HIV-1 RT by piperidine-substituted thiophene[3,2-*d*] pyrimidine NNRTIs

Our previous MT-4 cell-based antiviral activity evaluations showed that K-5a2 displays ~3-fold greater efficacy than ETR against the WT HIV-1 strain, and higher or similar efficacy against virus variants bearing four prevalent single-residue mutations (L100I, K103N, E138K and Y181C) in RT (hereinafter referred to as L100I RT, K103N RT, E138K RT and Y181C RT, respectively). However, K-5a2 is less effective than ETR in inhibiting HIV-1 strains containing K103N/Y181C RT (*Kang et al., 2016*). The compound 25a, resulting from further optimization of K-5a2, overcomes the limitations of K-5a2 and exhibits significantly better inhibitory effects on all tested HIV-1 strains (*Kang et al., 2017*). To better compare the anti-HIV-1 potency of 25a with that of existing NNRTIs, we measured the $EC_{50}$ values of RPV towards WT HIV-1 and mutant RT-bearing HIV-1 variants using the same method. 25a holds advantages over RPV in most of the tested drug-resistant HIV-1 strains while retaining similar antiviral potency against HIV-1 strains containing WT RT (*Table 1*). It is noteworthy that the particularly challenging K103N/Y181C double-mutation only causes a ~4.6-fold change in susceptibility to 25a, whereas it reduces the anti-HIV-1 efficacy of RPV by more than 10-fold. The superiority of 25a over RPV in targeting K103N/Y181C RT was further validated in the in vitro RT inhibition assay using purified recombinant RT variants, where K103N/Y181C mutation confers lower level of resistance to 25a (7.2-fold change in the $IC_{50}$ value) than to RPV (15-fold change) (*Figure 4A and B* and *Table 2*).

To further evaluate the resistance profile of 25a, we compared the RT inhibitory activities of 25a and RPV against two additional clinically relevant RT mutants, Y188L RT and V106A/F227L RT. While 25a and RPV exhibit similar inhibitory potency against WT RT, 25a is more resilient to Y188L and V106A/F227L mutations (0.70- and 1.7-fold change, respectively) than RPV (2.2- and 4.0-fold change, respectively) (*Figure 4A and B* and *Table 2*). To assess whether mutations against which RPV loses considerable potency would be susceptible to 25a, we tested the inhibitory activities of 25a and RPV against K101P RT, Y181I RT and K103N/Y181I RT, which were shown to cause substantial reduction in susceptibility to RPV (*Azijn et al., 2010*; *Giacobbi and Sluis-Cremer, 2017*; *Smith et al., 2016*). As expected, all three mutations dramatically lower the anti-RT potency of RPV and cause 20-fold, 90-fold and 1805-fold change in the $IC_{50}$ values, respectively. In contrast, there is considerably less resistance to 25a for all three RT mutants (1.3-, 8.8- and 96-fold, respectively) (*Figure 4C and D* and *Table 2*).

The longer right wing of 25a enables its interactions with NNIBP residues that are not contacted by RPV, such as Pro225 and Pro236. To gauge the likely impact of these mutations on 25a efficacy, we measured the RT-inhibiting potency of 25a against RT containing P225H or P236L substitutions,

**Table 1.** Anti-HIV-1 activity and cytotoxicity of K-5a2, 25a, etravirine (ETR) and rilpivirine (RPV) against wild-type (WT) HIV-1 and selected mutant HIV-1 strains in MT-4 cell assays.

| Inhibitor | | K-5a2[*] | 25a[†] | ETR[‡] | RPV[‡] |
|---|---|---|---|---|---|
| $EC_{50}$ (nM) | WT | 1.4 ± 0.43[§] | 1.2 ± 0.26 | 4.1 ± 0.15 | 0.99 ± 0.27 |
| | L100I | 3.4 ± 0.66 | 1.3 ± 0.50 | 5.4 ± 2.1 | 1.5 ± 0.0011 |
| | K103N | 2.9 ± 0.014 | 0.96 ± 0.07 | 2.4 ± 0.67 | 1.3 ± 0.36 |
| | E138K | 2.9 ± 0.021 | 4.7 ± 0.16 | 14 ± 2.3 | 5.7 ± 0.11 |
| | Y181C | 3.2 ± 0.48 | 5.0 ± 0.11 | 16 ± 2.1 | 5.0 ± 0.48 |
| | K103N/Y181C | 31 ± 12 | 5.5 ± 0.81 | 17 ± 1.8 | 11 ± 1.9 |
| $CC_{50}$ (µM) | | >227 | 2.3 ± 0.47 | >4.6 | 4.0 ± 1.2 |

* Results from (*Kang et al., 2016*).

† Results from (*Kang et al., 2017*).

‡ The data were obtained from the same laboratory using the same method.

§ Data reported as mean ± standard deviations.

DOI: https://doi.org/10.7554/eLife.36340.008

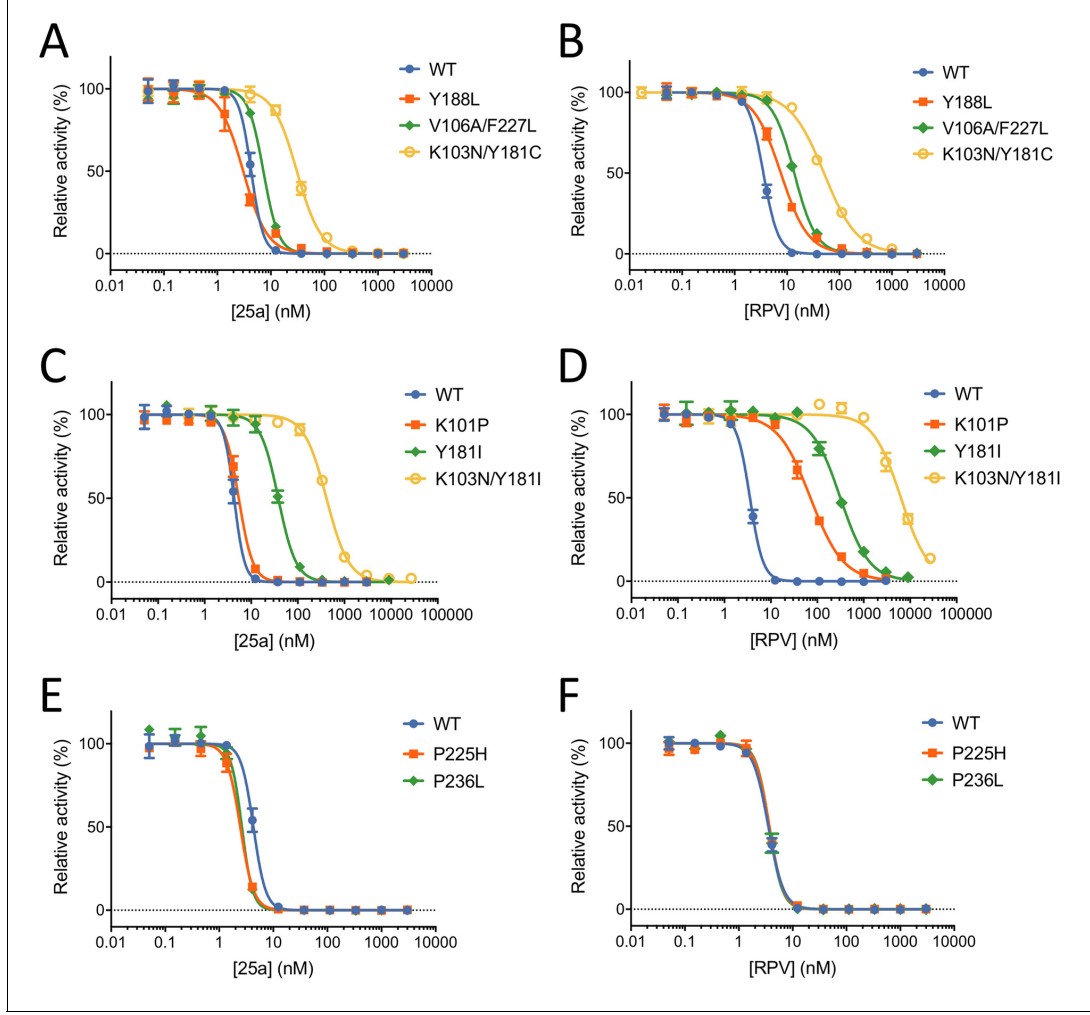

**Figure 4.** In vitro Inhibition of HIV-1 RT by 25a and RPV. (**A**) and (**B**) Inhibition curves of WT RT, Y188L RT, K103N/Y181C RT and V106A/F227L RT by 25a and RPV. (**C**) and (**D**) Inhibition curves of WT RT, K101P RT, Y181I RT and K103N/Y181I RT by 25a and RPV. (**E**) and (**F**) Inhibition curves of WT RT, P225H RT and P236L RT by 25a and RPV. Each data point is shown as mean ± standard error (n = 3). The data are fitted into inhibition dose-response curves with variable slopes. All datasets have excellent goodness of fit with $R^2 \geq 0.99$ except for the inhibition curve of RPV against K103N/Y181I RT ($R^2 = 0.98$). The $IC_{50}$ and curve slope values are summarized in *Table 2*.

DOI: https://doi.org/10.7554/eLife.36340.007

two clinically identified mutations shown no significant reduction in susceptibility to RPV (*Basson et al., 2015*). Like that of RPV, the potency of 25a was not negatively affected by either P225H or P236L mutations (0.58- and 0.60-fold change, respectively) (*Figure 4E and F* and *Table 2*), indicating that 25a has a relatively high genetic barrier to the development of novel drug-resistant mutations.

It is worth mentioning that 25a has steeper dose-response curve slopes than RPV in the inhibition of all above RT variants (*Table 2*). This characteristic can help 25a achieve greater inhibition of RT activity at higher than $IC_{50}$ concentrations, which are usually more clinically relevant (*Shen et al., 2008*). Taken together, by comparing the inhibitory potency of 25a and RPV in a wide range of RT mutants, we have shown that 25a has an improved resistance profile over RPV and is able to effectively inhibit the RT mutants causing high-level resistance to RPV.

**Table 2.** In vitro inhibition of HIV-1 reverse transcriptase by 25a and RPV.

| RT variants | 25a | | | RPV | | |
|---|---|---|---|---|---|---|
| | IC$_{50}$ (nM) | Fold R[*] | Curve slope | IC$_{50}$ (nM) | Fold R | Curve slope |
| WT | 4.3 ± 0.080[†] | – | 3.8 ± 0.70 | 3.5 ± 0.052 | – | 3.1 ± 0.17 |
| K103N/Y181C | 31 ± 0.83 | 7.2 | 1.9 ± 0.092 | 51 ± 1.5 | 15 | 1.4 ± 0.047 |
| Y188L | 3.0 ± 0.15 | 0.70 | 1.9 ± 0.16 | 7.6 ± 0.22 | 2.2 | 1.7 ± 0.069 |
| V106A/F227L | 7.3 ± 0.23 | 1.7 | 3.1 ± 0.17 | 14 ± 0.27 | 4.0 | 2.1 ± 0.083 |
| K101P | 5.4 ± 0.16 | 1.3 | 2.9 ± 0.21 | 71 ± 3.0 | 20 | 1.2 ± 0.056 |
| Y181I | 38 ± 1.1 | 8.8 | 2.3 ± 0.16 | 315 ± 15 | 90 | 1.4 ± 0.081 |
| K103N/Y181I | 412 ± 13 | 96 | 1.8 ± 0.094 | 6317 ± 339 | 1805 | 1.4 ± 0.10 |
| P225H | 2.5 ± 0.059 | 0.58 | 3.5 ± 0.15 | 3.7 ± 0.070 | 1.1 | 3.4 ± 0.34 |
| P236L | 2.6 ± 0.13 | 0.60 | 4.3 ± 0.43 | 3.7 ± 0.085 | 1.1 | 3.6 ± 0.47 |

[*] Mean fold change in the IC$_{50}$ values of mutant RT versus WT RT.

[†] Data reported as mean ±standard error.

DOI: https://doi.org/10.7554/eLife.36340.009

## Structural basis for improved resistance profile of piperidine-substituted thiophene[3,2-*d*]pyrimidine NNRTIs

To shed light on the mechanism underlying the outstanding resistance profile of the two piperidine-substituted thiophene[3,2-*d*]pyrimidine NNRTIs, we determined the crystal structures of K103N RT, E138K RT, and Y188L RT complexed with compound K-5a2, as well as K103N RT, E138K RT, K103N/Y181C RT, V106A/F227L RT, K101P RT, and Y181I RT complexed with 25a. The attempt to obtain the crystal structure of 25a in complex with K103N/Y181I RT proved unsuccessful, possibly due its suboptimal anti-RT potency towards K103N/Y181I RT, although it has displayed marked improvement over RPV in inhibiting this specific mutant (*Table 2*). Superposition of these mutant RT/NNRTI complexes structures onto their respective WT RT/NNRTI complexes structures shows no major deviation in the conformations of the enzyme and inhibitors (*Figure 5* and *Figure 5—figure supplement 1*). Root-mean-square deviations (RMSDs) for the structural alignments between WT RT/NNRTI complexes and mutant RT/NNRTI complexes range from 0.094 to 0.283 Å for the overall Cα atoms, and from 0.095 to 1.108 Å for the Cα atoms of the NNIBP regions (residues 98–110, 178–190, 226–240 of the p66 subunit, plus residues 137–139 of the p51 subunit) (*Table 3*).

Examination of the interactions between the RT mutants and the two NNRTIs reveals that all the hydrogen bonds depicted in *Figure 3D* are preserved, although there are some variations in the bond lengths. To analyze the extent of interactions between the inhibitors and different RT mutants, we measured the buried surface areas between the inhibitors and the whole NNIBP as well as a selection of key residues in the NNIBP of each RT variants (*Table 4*). In the structures of K103N RT in complex with 25a or K-5a2, the Lys to Asn substitution in RT shortens the aliphatic side chain and reduces the contact interface between residue 103 and the inhibitors, but 25a and K-5a2 are able to establish more contacts with Phe227 and Pro236 by varying their multiple torsion angles (*Table 5*) to counterbalance the loss (*Figure 5A* and *Figure 5—figure supplement 1A*). Similarly, in the structure of Y188L RT/K-5a2 complex, the cyano-dimethylphenyl group in K-5a2 is diverted away from Leu188 to avoid steric clashes, leading to declines in the buried areas between the inhibitor and Leu188 and Phe227. However, this mutation-caused damage is alleviated by its enhanced interactions with Lys103, Val106 and Pro236 (*Figure 5—figure supplement 1C*, *Table 4*). In the case of E138K RT, since the mutation does not disrupt the hydrophobic interactions between the inhibitors' central thiophene ring and residue 138, 25a and K-5a2 maintain almost the same binding poses in the NNIBP and similar buried areas with each of the residues along the pocket as in their complexes with WT RT (*Figure 5B* and *Figure 5—figure supplement 1B*).

In regard to RT carrying the more disruptive K103N/Y181C double-mutation, Y181C mutation abolishes the favorable π–π stacking interactions between the Tyr181 side chain and the dimethylphenyl ring of 25a, and greatly reduce the binding interface between 25a and Cys181. Moreover, the dramatic changes in the NNIBP result in a decrease of the buried interface between 25a and

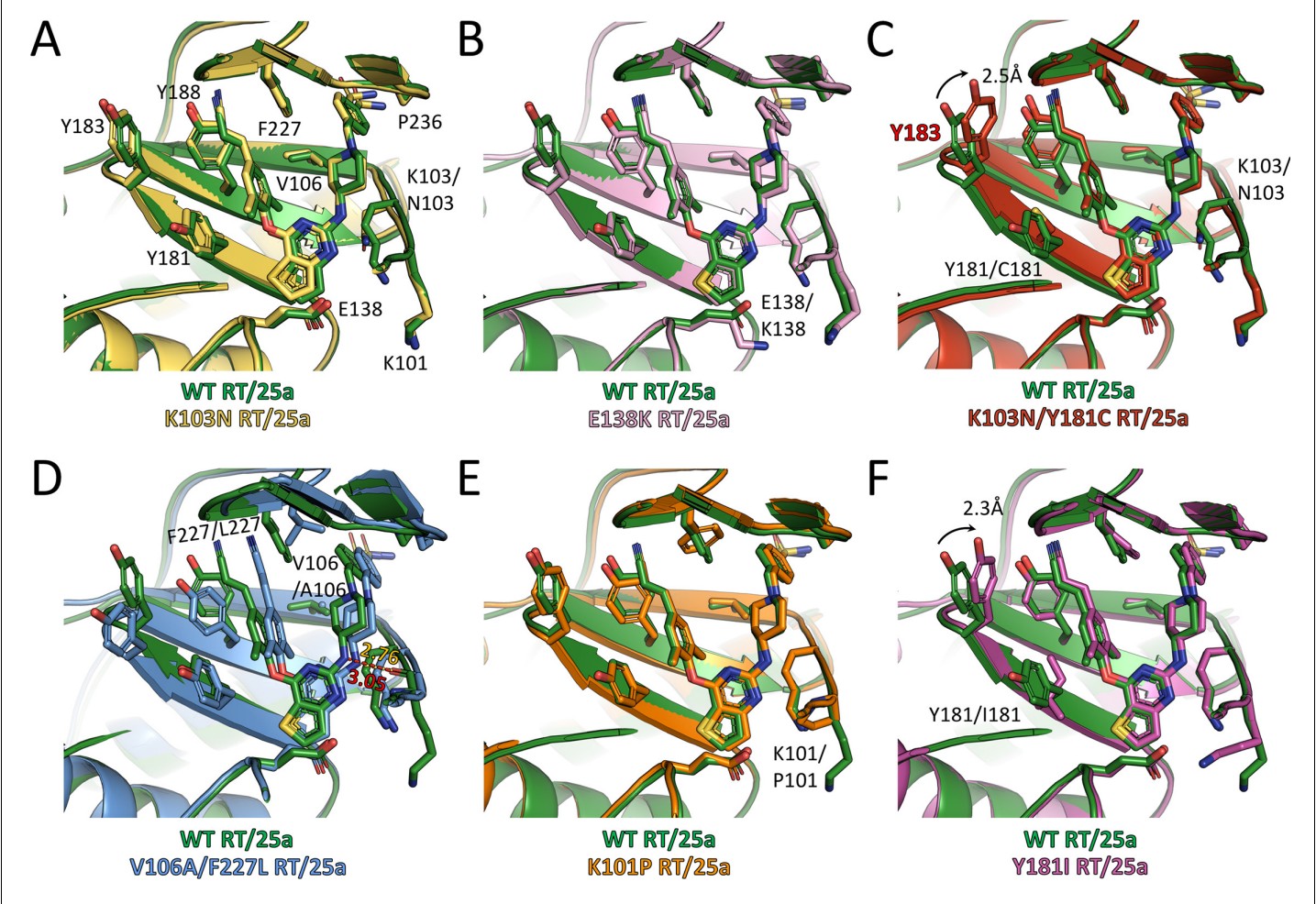

**Figure 5.** Comparison of the conformations of 25a in different RT complexes. Superposition of (**A**) K103N RT/25a complex structure, (**B**) E138K RT/25a complex structure, (**C**) K103N/Y181C RT/25a complex structure, (**D**) V106A/F227L RT/25a complex structure, (**E**) K101P RT/25a complex structure, and (**F**) Y181I RT/25a complex structure onto WT RT/25a complex structure. The structure of WT RT/25a complex is colored in green, K103N RT/25a complex in yellow, E138K RT/25a complex in pink, K103N/Y181C RT/25a complex in red, V106A/F227L RT/25a complex in blue, K101P RT/25a complex in orange, and Y181I RT/25a complex in magenta. Distances are in angstrom (Å).

DOI: https://doi.org/10.7554/eLife.36340.010

The following figure supplement is available for figure 5:

**Figure supplement 1.** Comparison of the conformations of K-5a2 and 25a in different RT complexes.

DOI: https://doi.org/10.7554/eLife.36340.011

residue 103. Nonetheless, the markedly weakened interactions between 25a and both mutated residues are remedied by the increase in the contact areas between 25a and several other residues in the NNIBP, including Tyr183, Phe227 and Pro236. (*Figure 5C*). In the V106A/F227L RT/25a complex structure, the much smaller side chain of Ala106 buries significantly less surface area with the inhibitor. Furthermore, the double-mutation causes more dramatic changes in the conformations of the NNIBP and the bound 25a. In particular, the cyanovinyl group of 25a is flipped so that it can maintain similar extent of interactions with the mutated Leu227. This torsional change, however, diverts the inhibitor away from the tunnel lined by Tyr181, Tyr183 and Tyr188 and diminishes the contact areas between 25a and all three tyrosine residues. To compensate for the loss, the right wing of 25a shifts closer to Lys101. Such movement shortens the distance between the linker amine group of 25a and the carboxyl oxygen of Lys101 from 3.05 to 2.76 Å and strengthens the hydrogen bond between them (*Figure 5D*). This hydrogen bond is conserved in the binding of many NNRTIs,

**Table 3.** Root mean square deviations (RMSDs) of Cα atoms (Å) for the alignments between different RT/NNRTI complexes structures.

|  | Overall | NNIBP region |
|---|---|---|
| WT RT/K-5a2 and K103N RT/K-5a2 | 0.283 | 0.579 |
| WT RT/K-5a2 and E138K RT/K-5a2 | 0.094 | 0.095 |
| WT RT/K-5a2 and Y188L RT/K-5a2 | 0.138 | 0.293 |
| WT RT/25a and K103N RT/25a | 0.250 | 0.430 |
| WT RT/25a and E138K RT/25a | 0.162 | 0.290 |
| WT RT/25a and K103N/Y181C RT/25a | 0.175 | 0.499 |
| WT RT/25a and V106A/F227L RT/25a | 0.245 | 1.108 |
| WT RT/25a and K101P RT/25a | 0.253 | 0.292 |
| WT RT/25a and Y181I RT/25a | 0.182 | 0.297 |

DOI: https://doi.org/10.7554/eLife.36340.012

including ETR and RPV, and contributes greatly to the binding affinities of NNRTIs (*Das et al., 2008*; *Lansdon et al., 2010*).

With respect to K101P RT, the deprotonation of Pro101 main-chain nitrogen attenuates its water-mediated hydrogen bond with the pyrimidine nitrogen in 25a. Nonetheless, the Lys to Pro substitution places its cyclic side chain in the vicinity of the central thiophene pyrimidine ring of 25a and leads to enhanced hydrophobic interactions (*Figure 5E*). As to the Y181I RT/25a complex structure, the mutation not only removes the π-π stacking interactions in the left wing of 25a, but also introduces steric hinderance with the linker oxygen and thiophene sulfur group in 25a, which pushes the inhibitor slightly away from the tunnel. This unfavorable change in the NNIBP is mitigated by the enlarged contact areas between 25a and the more closely placed side-chains of Lys101 and Tyr183 (*Figure 5F*).

It is noteworthy that in both K103N/Y181C RT/25a and Y181I RT/25a complex structures, Tyr183, a residue in the conserved YMDD motif at the polymerase active site, moves 2.3–2.5 Å towards the inhibitor, enhancing its hydrophobic interactions with the cyanovinyl group of 25a. The recruitment of Tyr183 by the cyanovinyl group of 25a is reminiscent of that observed in the structure of K103N/Y181C RT in complex with RPV, whose left wing has the same 4-cyanovinyl-2,6-dimethylphenyl structure (*Das et al., 2008*). Interestingly, in the structure of K103N RT/K-5a2 complex, Tyr183 also undergoes a significant conformational change and is placed closer to the inhibitor, although it is still

**Table 4.** Buried area (Å$^2$) between HIV-1 RT and each NNRTI[*].

|  | Total | Individual residue | | | | | | | | |
|---|---|---|---|---|---|---|---|---|---|---|
|  |  | 101 | 103 | 106 | 181 | 183 | 188 | 227 | 236 | 138 |
| WT RT/K-5a2 | 584.1 | 46.5 | 81.2 | 91.1 | 63.9 | 0 | 75.4 | 89.3 | 69.6 | 54.6 |
| K103N RT/K-5a2 | 579.4 | 44.3 | 73.4 | 85.5 | 59.6 | 0 | 71.2 | 92.9 | 71.2 | 52.6 |
| E138K RT/K-5a2 | 587.9 | 44.3 | 83.5 | 89.6 | 63.0 | 0 | 72.9 | 86.4 | 69.9 | 54.3 |
| Y188L RT/K-5a2 | 571.3 | 44.3 | 83.0 | 93.9 | 57.9 | 0 | 70.0 | 73.2 | 74.0 | 51.5 |
| WT RT/25a | 620.5 | 51.9 | 88.2 | 91.7 | 68.5 | 19.6 | 84.8 | 96.9 | 68.4 | 54.8 |
| K103N RT/25a | 626.8 | 47.9 | 74.7 | 86.4 | 67.8 | 15.6 | 84.3 | 106 | 72.0 | 53.7 |
| E138K RT/25a | 621.4 | 45.5 | 87.1 | 92.5 | 66.4 | 14.0 | 86.6 | 98.4 | 69.1 | 54.1 |
| K103N/Y181C RT/25a | 624.7 | 51.9 | 76.1 | 86.8 | 56.5 | 33.5 | 83.2 | 106 | 73.3 | 54.1 |
| V106A/F227L RT/25a | 614.9 | 47.6 | 87.1 | 79.2 | 58.1 | 0 | 79.2 | 91.8 | 62.8 | 52.0 |
| K101P RT/25a | 613.6 | 62.6 | 93.9 | 90.7 | 66.6 | 21.1 | 85.6 | 89.9 | 70.5 | 53.5 |
| Y181I RT/25a | 618.5 | 62.9 | 90.9 | 90.8 | 58.9 | 31.5 | 84.6 | 99.8 | 71.7 | 52.4 |

[*] The buried area between HIV-1 RT and each NNRTI was calculated using UCSF ChimeraX.
DOI: https://doi.org/10.7554/eLife.36340.013

**Table 5.** Torsion angles and energies of K-5a2 and 25a in different binding poses.

| | Torsion angles (°) | | | | | | | | NNRTI energy* (kcal/mol) |
|---|---|---|---|---|---|---|---|---|---|
| | τ1 | τ2 | τ3 | τ4 | τ5 | τ6 | τ7 | τ8 | |
| WT RT/K-5a2 | 14 | −17 | −84 | −71 | 2 | −8 | −97 | — | −161.4 |
| K103N RT/K-5a2 | 11 | −23 | −83 | −73 | 6 | −11 | −102 | — | −162.4 |
| E138K RT/K-5a2 | 9 | −19 | −87 | −68 | 2 | −9 | −97 | — | −162.7 |
| Y188L RT/K-5a2 | 3 | −49 | −66 | −70 | 4 | −2 | −94 | — | −160.9 |
| WT RT/ETR† | — | — | — | 16 | −2 | −13 | −95 | — | N/A |
| WT RT/25a | 23 | −26 | −79 | −79 | 1 | 0 | −109 | −40 | −191.1 |
| K103N RT/25a | 7 | −22 | −81 | −74 | 3 | 0 | −110 | −53 | −187.7 |
| E138K RT/25a | 19 | −29 | −76 | −77 | 0 | 3 | −108 | −53 | −188.5 |
| K103N/Y181C RT/25a | −4 | −26 | −76 | −77 | 4 | −1 | −107 | −54 | −186.2 |
| V106A/F227L RT/25a | 14 | −24 | −72 | −82 | 4 | −4 | −103 | −163 | −187.3 |
| K101P RT/25a | 5 | −24 | −84 | −81 | 5 | 6 | −112 | −44 | −190.1 |
| Y181I RT/25a | −2 | −15 | −87 | −72 | 6 | 4 | −107 | −57 | −189.2 |
| WT RT/RPV‡ | — | — | — | 10 | −7 | −13 | −103 | −28 | N/A |

\* The NNRTI energy refers to the energy of K-5a2 or 25a itself at the specific conformations in different RT complexes. It was calculated using the Macro-Model program in the Schrödinger software suite.

† Torsion angles of ETR were measured using the structure from PDB ID: 3MEC.

‡ Torsion angles of RPV were measured using the structure from PDB ID: 4G1Q.

DOI: https://doi.org/10.7554/eLife.36340.014

outside the contact radius of the cyano group in K-5a2. Superposition of the structures of WT RT/K-5a2, K103N RT/K-5a2, WT RT/25a, and K103N/Y181C RT/25a complexes reveals a gradual rotation of Tyr183 from the 'down' position in WT RT/K-5a2 complex to the 'up' position in K103N/Y181C RT/25a complex (*Figure 5—figure supplement 1D*). This stepwise movement of Tyr183 is likely triggered by three factors: (i) inhibitor repositioning because of the K103N mutation, (ii) loss of aromatic interactions due to the Y181C or Y181I mutation, and (iii) presence of a cyanovinyl group in the inhibitor left wing. Tyr183 makes the most significant contribution to the NNRTI-binding in the circumstance that all of the above factors are present. The ability of K-5a2 and 25a to recruit Tyr183 is particularly significant for their function of inhibiting the polymerase activity, because Tyr183 is completely conserved among all HIV-1 sequences and makes direct contacts with the nucleic acid substrate (*Das et al., 2012*; *Sarafianos et al., 2001*). The repositioning of Tyr183 towards the NNIBP removes this important interaction and destabilizes the binding of nucleic acid.

## Comparison of the binding modes for K-5a2, 25a and DAPY NNRTIs

By adopting the typical horseshoe conformation, K-5a2 and 25a substantially overlap the binding sites occupied by ETR and RPV (*Figure 6*). The thiophene substituent in the central ring of K-5a2 and 25a extends further into the entrance channel and is proximal to Glu138 located at the opening. The positions of their left wing structures are adjusted to small conformational changes of Tyr181, Tyr183 and Tyr188 to maximize the contacts with the pocket residues in this region. The piperidine ring in the right wing of K-5a2 and 25a slants away from the cyanophenyl plane in ETR and RPV through a ~ 90° tilt of τ4 (from 16°/10° to –71°/–79°) (*Table 5*) (*Das et al., 2008*; *Lansdon et al., 2010*), leading to the displacement of Tyr318 ~1 Å away from the inhibitors. The main difference in RT conformation is the uplift of the loop proceeding β9 and that connecting β10–β11. Upon the binding of K-5a2 and 25a, Pro225 and Pro236, two residues sitting at the groove channel opening, are pushed apart to accommodate the benzenesulfonamide group, which protrudes to the solvent-exposed surface of the enzyme.

There is no structure available for E138K RT and K101P RT in complex with either ETR or RPV; however, the structures of these two RT mutants in complex with K-5a2 and 25a provide a structural basis to understand why the mutations can render ETR and RPV less effective. In the structure of WT RT/ETR complex, the amino substituent of the central pyrimidine ring forms a salt bridge with the

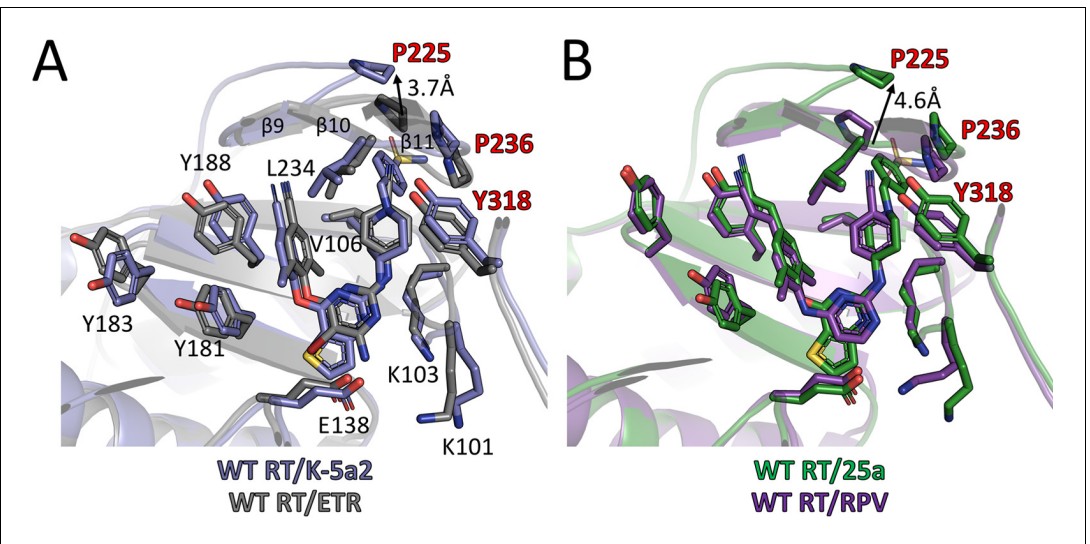

**Figure 6.** Comparison of the binding modes for K-5a2, 25a, ETR and RPV in RT. (**A**) Superposition of WT RT/ETR complex structure (PDB ID: 3MEC) onto WT RT/K-5a2 complex structure. The structure of WT RT/K-5a2 complex is colored in blue and WT RT/ETR complex in gray. (**B**) Superposition of WT RT/RPV complex structure (PDB ID: 4G1Q) onto WT RT/25a complex structure. The structure of WT RT/25a complex is colored in green and WT RT/RPV complex in purple.

DOI: https://doi.org/10.7554/eLife.36340.015

The following figure supplement is available for figure 6:

**Figure supplement 1.** Mechanisms of resistance to ETR and RPV in E138K RT and K101P RT.

DOI: https://doi.org/10.7554/eLife.36340.016

carboxyl side chain of Glu138. Transplanting Lys138 from the structure of E138K RT/K-5a2 complex into this structure reveals a severe charge-charge repulsion between the amino group in ETR and the side chain of Lys138, which would destabilize the binding of the inhibitor (*Figure 6—figure supplement 1A*). In the case of PRV, Glu138 contributes to the RT-RPV interactions by bonding Lys101 and placing it in the vicinity of the central pyrimidine ring for several critical interactions: (i) hydrophobic interactions between the pyrimidine ring of RPV and the Cε atom of Lys101, (ii) the hydrogen bond between the pyrimidine nitrogen atom and the main-chain nitrogen of Lys101, and (iii) the hydrogen bond between a linker nitrogen atom of RPV and the carbonyl oxygen of Lys101. Modeling Lys138 from the structure of E138K RT/25a complex indicates that Lys at residue 138 of the p51 subunit would shove Lys101 away from RPV. This conformational change in RT would not only disrupt the hydrophobic interactions but also weaken the two key hydrogen bonds due to the deviation of the Lys101 backbone (*Figure 6—figure supplement 1B*).

The K101P mutation considerably reduces susceptibility to RPV. Superposition of the WT RT/RPV and K101P RT/25a complex structures reveals that the mutation would remove the hydrophobic interaction between the RPV pyrimidine ring and the long aliphatic side chain of Lys101, and possibly introduces steric clashes between the polar groups in the pyrimidine ring and the non-polar side chain of Pro101. More importantly, Lys to Pro substitution will abrogate the critical hydrogen bond between the pyrimidine nitrogen of RPV and the backbone nitrogen in RT due to deprotonated form of Pro101 main-chain nitrogen and hence its inability to act as a hydrogen bond donor (*Figure 6—figure supplement 1C*).

The improved resistance profiles of 25a over RPV against other RT mutants, especially Y181I RT, Y188L RT and K103N/Y181C RT, are likely due to the bulkier substituents in its right wing and more extensive hydrogen bond interactions with NNIBP residues. Therefore, the π-π interactions provided by Tyr181 and Tyr188 likely make a much less contribution to the binding affinity of 25a than to that of RPV. Moreover, the higher degree of strategic flexibility of 25a (because of more rotatable bonds)

can make it more effective to preserve or even enhance its interactions with other NNIBP residues when Y181I and Y188L mutations displace the left wing structure of 25a.

## Discussion

Emergence of drug-resistant mutations in HIV-1 RT remains a major challenge for the design and development of NNRTIs. Using ETR as a lead compound, our previous efforts led to the design of two piperidine-substituted thiophene[3,2-*d*]pyrimidine derivatives, K-5a2 and 25a, with single-digit nanomolar $EC_{50}$ values against HIV-1 strains containing either WT RT or RT variants bearing various resistance-associated mutations. In the current study, we have shown that 25a is more effective than RPV against a broad set of RT mutants and determined the crystal structures of both WT RT and a number of RT mutants in complex with either K-5a2 or 25a. These high-resolution structures enable unambiguous determination of the binding modes of K-5a2 and 25a, and accurate illustration of the detailed interactions between RT and these highly potent NNRTIs. By virtue of their structural flexibility, K-5a2 and 25a are able to adapt to the conformational changes of RT induced by mutations in the NNIBP and optimize their complementarity with the mutated pocket by varying their multiple torsion angles. As such, the buried areas between the inhibitors and RT are similar across WT RT and various RT mutants, suggesting K-5a2 and 25a can occupy the NNIBP of RT mutants as effectively as they bind to the pocket of WT RT (*Table 4*). Energy calculation of the K-5a2 and 25a shows that the NNRTIs in different RT-bound conformations are almost isoenergetic (*Table 5*), indicating the conformational changes of K-5a2 and 25a induced by NNIBP residue mutations do not bring in significant strain energy penalties. Although in cellular environment, both RT and the bound inhibitors are in constant motion, and the interactions between them are undergoing repeated breaking and re-forming, the binding mode captured in the crystal structure should represent the averaged state of the complex or a state with high likelihood.

Aside from structural flexibility, hydrogen bonding with the main chains of NNIBP residues was previously suggested as another method to design NNRTIs that can overcome the effects of drug-resistant mutations in RT (*Zhan et al., 2009*). Compared with ETR and RPV, K-5a2 and 25a form considerably more hydrogen bonds between their polar groups (the thiophene sulfur, piperidine nitrogen and solvent-exposed sulfonamide) and the main chains of residues throughout the binding pocket. This extensive network of main-chain hydrogen bonds contributes substantially to the free energy of RT-inhibitor binding and is less susceptible to side chain mutations in the pocket. Furthermore, the more extended right wing structures of K-5a2 and 25a contact with a larger set of NNIBP residues than DAPY NNRTIs. Although it potentially makes K-5a2 and 25a susceptible to mutations of the residues not contacted by ETR or RPV, our results have shown that mutations of Pro225 and Pro236, whose side chains interact with the right wing of 25a, but not that of RPV, do not cause resistance to 25a. Additional RT-25a interactions that are not present in the RT/RPV complex include the hydrogen bonds between the sulfonamide group of 25a and Lys104 and Val106. The mutations of these two residues are unlikely to cause loss of potency of 25a because the two hydrogen bonds are established through the main chains of Lys104 and Val106. Even if the side-chain substitutions deviate the main chains, a minor change in the torsion angle $\tau 1$, which was shown to span a wide range without significant energetic penalty (*Table 5*), would readily place the sulfonamide group of 25a in an optimal position for hydrogen bonds formation.

In conclusion, our study depicts the binding poses of two newly developed NNRTIs, compounds K-5a2 and 25a, in their complexes with WT and mutant RTs, and exemplifies how broadly active NNRTIs retain satisfactory activities against RT containing drug-resistant mutations by taking advantage of the plasticity of both the inhibitors and the NNIBP of RT. Our findings provide a reliable model to analyze the structural effects of drug-resistant mutations in RT, and will contribute to structure-based design of novel NNRTIs that can effectively target multiple variants of RT.

## Materials and methods

**Key resources table**

| Reagent type (species) or resource | Designation | Source or reference | Identifiers | Additional information |
|---|---|---|---|---|

*Continued on next page*

*Continued*

| Reagent type (species) or resource | Designation | Source or reference | Identifiers | Additional information |
|---|---|---|---|---|
| Strain, strain background (*E.coli*) | BL21 Star (DE3) | ThermoFisher Scientific | C601003 | Chemically Competent *E.coli* for the expression of recombinant RTs. |
| Strain, strain background (HIV-1 IIIB) | HIV-1 L100I | Established in house | | |
| Strain, strain background (HIV-1 IIIB) | HIV-1 K103N | Established in house | | |
| Strain, strain background (HIV-1 IIIB) | HIV-1 E138K | Established in house | | |
| Strain, strain background (HIV-1 IIIB) | HIV-1 Y181C | Established in house | | |
| Strain, strain background (HIV-1 IIIB) | HIV-1 K103N/Y181C | Established in house | | |
| Cell line (*H. Sapiens*) | MT-4 cells | NIH AIDS Reagent Program | NIH-ARP Cat# 120–438, RRID:CVCL_2632 | |
| Recombinant DNA reagent | pCDFDuet-1 | Millipore Sigma | 71340–3 | Expression plasmid for all RT variants in *E.coli*. |
| Peptide, recombinant protein | HRV 3C protease | Recombinantly expressed in house | | Expressed as His-tagged fusion protein. |
| Commercial assay or kit | EnzChek Reverse Transcriptase Assay kit | ThermoFisher Scientific | E22064 | |
| Chemical compound, drug | Compound K-5a2 | Synthesized in house | | |
| Chemical compound, drug | Compound 25a | Synthesized in house | | |
| Chemical compound, drug | etravirine | Sigma-Aldrich | ADV428293567 | |
| Chemical compound, drug | rilpivirine | Sigma-Aldrich | ADV465749297 | |
| Chemical compound, drug | PicoGreen dsDNA reagent | ThermoFisher Scientific | P7581 | |
| Software, algorithm | *Coot* | (*Emsley et al., 2010*) | RRID:SCR_014222 | |
| Software, algorithm | XDS | (*Kabsch, 2010*) | RRID:SCR_015652 | |
| Software, algorithm | PHASER | (*McCoy et al., 2007*) | RRID:SCR_014219 | |
| Software, algorithm | PHENIX suite | (*Adams et al., 2010*) | RRID:SCR_014224 | |
| Software, algorithm | PyMol v 2.0 | The PyMOL Molecular Graphics System, Schrödinger, LLC. | RRID:SCR_000305 | |
| Software, algorithm | MacroModel | Schrödinger suite | RRID:SCR_014879 | |
| Software, algorithm | UCSF Chimera | (*Pettersen et al., 2004*) | RRID:SCR_004097 | |
| Software, algorithm | UCSF ChimeraX | (*Goddard et al., 2018*) | RRID:SCR_015872 | |
| Software, algorithm | GraphPad Prism v 7.0a | GraphPad Software | RRID:SCR_002798 | |

## Cloning, protein preparation and crystallization

An engineered HIV-1 RT construct, RT52A (*Bauman et al., 2008*; *Das et al., 2008*), here referred to as WT RT, was used as the template for site-directed mutagenesis to introduce E138K mutation in the p51 subunit, K101P, K103N, Y181I, Y188L, K103N/Y181C, K103N/Y181I and V106A/F227L mutations in the p66 subunit. WT and mutant RTs were expressed and purified as described previously (*Das et al., 2008*; *Frey et al., 2015*). Briefly, the p51 subunit with an N-terminal 6xHis tag followed by a human rhinovirus (HRV) 3C protease cleavage site and un-tagged p66 subunit were co-expressed in *E. coli* BL21 star (DE3) (Thermo Fisher Scientific, Waltham, MA). Cells were grown at 37°C and induced at 17°C for 16 hr. WT and mutant RTs were purified on a HisTrap affinity column and a HiTrap Heparin affinity column (GE Healthcare), sequentially. The N-terminal 6xHis tag was removed by HRV 3C protease, and the un-tagged RT was purified on a Superdex 200 gel filtration column (GE Healthcare) in buffer containing 10 mM Tris (pH 8.0), 75 mM NaCl and 2 mM Tris(2-carboxyethyl)phosphine (TCEP).

Crystallization of WT and mutant RTs were set up using the sitting drop vapor diffusion method at 4°C, with 2 μl of protein solution added to 2 μl of well buffer containing 50 mM MES or imidazole buffer (pH 6.0–6.6), 10% (v/v) polyethylene glycol (PEG) 8000, 100 mM ammonium sulfate, 15 mM magnesium sulfate, and 10 mM spermine. Crystals were grown for 2 weeks, and RT/NNRTI complexes were prepared by soaking RT crystals in buffer containing 0.5 mM K-5a2 or 25a, 50 mM MES

or imidazole buffer (pH 6.0), 12% (v/v) polyethylene glycol (PEG) 8000, 100 mM ammonium sulfate, 15 mM magnesium sulfate, 10 mM spermine, 25% ethylene glycol, and 10% DMSO for 2 days. Soaked crystals were harvested and flash-frozen in liquid nitrogen.

## Data collection and structure determination

X-ray diffraction data were collected at the Advanced Photon Source at Argonne National Laboratory on beamline 24ID-E at a wavelength of 0.97918 Å. Data sets were integrated and scaled with XDS software package (*Kabsch, 2010*). Structures of RT/K-5a2 and RT/25a complexes were determined by molecular replacement in Phaser (*McCoy et al., 2007*) using the structure of WT RT/RPV complex (PDB ID: 4G1Q) as a search template. One RT molecule was present in the asymmetric unit. The ligand restraints and 3D structures of K-5a2 and 25a were generated in eLBOW (*Moriarty et al., 2009*) using SMILES strings as inputs. Models of K-5a2 and 25a were built into the structures based on the unbiased $F_o - F_c$ difference Fourier electron density map calculated in the absence of an NNRTI. Models were manually rebuilt in *Coot* (*Emsley et al., 2010*), and refined in PHENIX (*Adams et al., 2010*). The quality of the final models was analyzed with MolProbity (*Chen et al., 2010*). Data collection and refinement statistics are summarized in *Supplementary file 1*. All figures were generated using PyMOL, UCSF Chimera (*Pettersen et al., 2004*) or UCSF ChimeraX (*Goddard et al., 2018*).

## Cell lines

MT-4 cells were obtained from the NIH AIDS Reagent Program and authenticated by the supplier. All cells are tested negative for mycoplasma, bacteria, and fungi.

## T cell-based anti-HIV-1 activity assays

The anti-HIV-1 activities of rilpivirine (RPV) against WT HIV-1 (IIIB strain) as well as seven mutant RT-carrying HIV-1 variants (L100I, K103N, E138K, Y181C and K103N/Y181C) were evaluated in MT-4 cells using MTT method as described previously (*Kang et al., 2017*, *2016*; *Pannecouque et al., 2008*). Briefly, stock solutions (10 × final concentration) of RPV were added in 25 µl to two series of triplicate wells in order to allow simultaneous evaluation of their effects on mock- and HIV-1-infected cells. Using a Biomek 3000 robot (Beckman Instruments, Fullerton, CA), nine five-fold serial dilutions of RPV (final 200 µl volume per well) were made directly in flat-bottomed 96-well microtiter trays, including untreated control HIV-1 and mock-infected cell samples for each sample. Stock of WT HIV-1 or mutant HIV-1 strains (50 µl at 100–300-fold of 50% cell culture infectious dose (CCID$_{50}$)) or equal amount of culture medium was added to either HIV-1-infected or mock-infected wells of the microtiter tray. Mock-infected cells were used to evaluate the cytotoxicity of the compounds. Exponentially growing MT-4 cells were centrifuged for 5 min at 220 × g and the supernatant was discarded. The MT-4 cells were resuspended at $6 \times 10^5$ cells/ml, and 50 µl aliquots were transferred to the microtiter tray wells. At five days after infection, the viability of mock- and HIV-1-infected cells was determined spectrophotometrically in an Infinite M1000 microplate reader (Tecan, Zürich, Switzerland). All data were calculated using the median optical density (OD) value of triplicate wells. The 50% effective antiviral concentration (EC$_{50}$) was defined as the concentration of the test compound affording 50% protection from viral cytopathogenicity. The 50% cytotoxic concentration (CC$_{50}$) was defined as the compound concentration that reduced the absorbance (OD$_{540\ nm}$) of mock-infected MT-4 cells by 50%. The results are presented as mean ± SD (n = 3).

## Reverse transcriptase inhibition assays

The HIV-1 RT inhibition assay was performed using a PicoGreen-based EnzChek Reverse Transcriptase Assay kit (Thermo Fisher Scientific) according to manufacturer's protocol with minor modifications. Briefly, 58 µl of Recombinant WT or mutant RT (final concentration in reaction is 20 nM) in buffer containing 50 mM Tris (pH 8.0), 50 mM KCl, 6 mM MgCl$_2$, and 10 mM DTT was incubated with 2 µl 25a or RPV (Sigma-Aldrich) with a concentration gradient comprising eleven three-fold serial dilutions of each inhibitor, or equal amount of DMSO at 25°C for 1 hr. 30 µl of pre-annealed poly(rA)•d(T)$_{16}$ in buffer containing 50 mM Tris (pH 8.0), 50 mM KCl, 6 mM MgCl$_2$, 10 mM DTT, and 100 mM dTTP was added to the RT-inhibitor mixture to start DNA polymerization reaction. After 30 min of incubation at 25°C, 10 µl of 150 mM EDTA was added to stop the reaction. 100 µl of 2x

PicoGreen reagent was then added to each reaction and product formation was quantified using a TriStar LB 941 microplate reader (Berthold Technologies) with excitation/emission = 485/535 nm. The activity of WT or each mutant RT in the presence of inhibitors was normalized against the DMSO control. The $IC_{50}$ and curve slope values were calculated by fitting the data into inhibition dose-response curves with variable slopes using GraphPad Prism version 7.0a. The experiment was repeated three times independently.

## Accession numbers

The atomic coordinates and structure factors have been deposited in the Protein Data Bank under the accession codes 6C0J, 6C0K, 6C0L, 6CGF, 6C0N, 6C0O, 6C0P, 6C0R, 6DUF, 6DUG, and 6DUH.

## Acknowledgements

We thank Dr. Stefan Sarafianos and Dr. Karen Anderson for the HIV-1 RT52A construct, Dr. Chang Liu and Dr. Bin Liu for advice and critical reading of the manuscript. We thank the staff of the Advanced Photon Source beamline 24-ID-E for technical assistance with data collection and the Richards Center at Yale University for computational support. This work was supported by Howard Hughes Medical Institute and National Institutes of Health (GM022778 to T.A.S.); National Natural Science Foundation of China (81273354 to X.L.); Key Project of National Natural Science Foundation of China for International Cooperation (81420108027 to X.L.); Key research and development project of Shandong Province (2017CXGC1401 to X.L.); Major Project of Science and Technology of Shandong Province (2015ZDJS04001 to X.L.); Young Scholars Program of Shandong University (2016WLJH32 to P.Z.).

## Additional information

### Funding

| Funder | Grant reference number | Author |
| --- | --- | --- |
| Howard Hughes Medical Institute | Investigator Program | Thomas A Steitz |
| National Institute of General Medical Sciences | GM022778 | Thomas A Steitz |
| National Natural Science Foundation of China | 81273354 | Xinyong Liu |
| Shandong Province | Key Research and Development Project: 2017CXGC1401 | Xinyong Liu |
| Shandong Province | Major Project of Science and Technology: 2015ZDJS04001 | Xinyong Liu |
| Shandong University | Young Scholars Program: 2016WLJH32 | Peng Zhan |
| National Natural Science Foundation of China | Key Project for International Cooperation: 81420108027 | Xinyong Liu |

The funders had no role in study design, data collection and interpretation, or the decision to submit the work for publication.

### Author contributions

Yang Yang, Conceptualization, Data curation, Formal analysis, Validation, Investigation, Visualization, Methodology, Writing—original draft, Writing—review and editing; Dongwei Kang, Data curation, Formal analysis, Validation, Investigation, Methodology, Writing—review and editing; Laura A Nguyen, Conceptualization, Data curation, Formal analysis, Investigation, Methodology, Writing—review and editing; Zachary B Smithline, Christophe Pannecouque, Data curation, Formal analysis, Investigation, Writing—review and editing; Peng Zhan, Data curation, Formal analysis, Funding

acquisition, Investigation, Methodology, Writing—review and editing; Xinyong Liu, Resources, Supervision, Funding acquisition, Writing—review and editing; Thomas A Steitz, Resources, Supervision, Funding acquisition, Validation, Project administration, Writing—review and editing

### Author ORCIDs
Yang Yang  http://orcid.org/0000-0001-9061-3828
Dongwei Kang  https://orcid.org/0000-0001-9232-953X
Thomas A Steitz  http://orcid.org/0000-0002-3357-3505

### Decision letter and Author response
Decision letter https://doi.org/10.7554/eLife.36340.046
Author response https://doi.org/10.7554/eLife.36340.047

## Additional files

### Supplementary files
• Supplementary file 1. X-ray crystallography statistics from data collection and refinement.
DOI: https://doi.org/10.7554/eLife.36340.017
• Transparent reporting form
DOI: https://doi.org/10.7554/eLife.36340.018

### Data availability
Diffraction data and atomic coordinates have been deposited in the Protein Data Bank under the accession codes 6C0J, 6C0K, 6C0L, 6CGF,6C0N, 6C0O, 6C0P, 6C0R, 6DUF, 6DUG, and 6DUH.

The following datasets were generated:

| Author(s) | Year | Dataset title | Dataset URL | Database, license, and accessibility information |
|---|---|---|---|---|
| Yang Y, Nguyen AL, Smithline ZB, Steitz TA | 2018 | Crystal structure of HIV-1 reverse transcriptase in complex with nonnucleoside inhibitor K-5a2 | https://www.rcsb.org/structure/6c0j | Publicly available at the RCSB Protein Data Bank (accession no. 6C0J). |
| Yang Y, Nguyen AL, Smithline ZB, Steitz TA | 2018 | Crystal structure of HIV-1 K103N mutant reverse transcriptase in complex with non-nucleoside inhibitor K-5a2 | https://www.rcsb.org/structure/6C0K | Publicly available at the RCSB Protein Data Bank (accession no. 6C0K). |
| Yang Y, Nguyen AL, Smithline ZB, Steitz TA | 2018 | Crystal structure of HIV-1 E138K mutant reverse transcriptase in complex with non-nucleoside inhibitor K-5a2 | https://www.rcsb.org/structure/6c0l | Publicly available at the RCSB Protein Data Bank (accession no. 6C0L). |
| Yang Y, Nguyen AL, Smithline ZB, Steitz TA | 2018 | Crystal structure of HIV-1 Y188L mutant reverse transcriptase in complex with non-nucleoside inhibitor K-5a2 | https://www.rcsb.org/structure/6cgf | Publicly available at the RCSB Protein Data Bank (accession no. 6CGF). |
| Yang Y, Nguyen AL, Smithline ZB, Steitz TA | 2018 | Crystal structure of HIV-1 reverse transcriptase in complex with non-nucleoside inhibitor 25a | https://www.rcsb.org/structure/6c0n | Publicly available at the RCSB Protein Data Bank (accession no. 6C0N). |
| Yang Y, Nguyen AL, Smithline ZB, Steitz TA | 2018 | Crystal structure of HIV-1 K103N mutant reverse transcriptase in complex with non-nucleoside inhibitor 25a | https://www.rcsb.org/structure/6c0o | Publicly available at the RCSB Protein Data Bank (accession no. 6C0O). |
| Yang Y, Nguyen AL, Smithline ZB, Steitz TA | 2018 | Crystal structure of HIV-1 E138K mutant reverse transcriptase in complex with non-nucleoside inhibitor 25a | https://www.rcsb.org/structure/6c0p | Publicly available at the RCSB Protein Data Bank (accession no. 6C0P). |
| Yang Y, Nguyen AL, Smithline ZB, Steitz TA | 2018 | Crystal structure of HIV-1 K103N/Y181C mutant reverse transcriptase in complex with non-nucleoside | https://www.rcsb.org/structure/6c0r | Publicly available at the RCSB Protein Data Bank (accession |

| Author(s) | Year | Dataset title | Dataset URL | Database, license, and accessibility information |
|---|---|---|---|---|
| | | inhibitor 25a | | no. 6C0R). |
| Yang Y, Nguyen AL, Smithline ZB, Steitz TA | 2018 | Crystal structure of HIV-1 reverse transcriptase V106A/F227L mutant in complex with non-nucleoside inhibitor 25a | https://www.rcsb.org/structure/6duf | Publicly available at the RCSB Protein Data Bank (accession no. 6DUF). |
| Yang Y, Nguyen AL, Smithline ZB, Steitz TA | 2018 | Crystal structure of HIV-1 reverse transcriptase K101P mutant in complex with non-nucleoside inhibitor 25a | https://www.rcsb.org/structure/6dug | Publicly available at the RCSB Protein Data Bank (accession no. 6DUG). |
| Yang Y, Nguyen AL, Smithline ZB, Steitz TA | 2018 | Crystal structure of HIV-1 reverse transcriptase Y181I mutant in complex with non-nucleoside inhibitor 25a | https://www.rcsb.org/structure/6duh | Publicly available at the RCSB Protein Data Bank (accession no. 6DUH). |

The following previously published datasets were used:

| Author(s) | Year | Dataset title | Dataset URL | Database, license, and accessibility information |
|---|---|---|---|---|
| Bauman JD, Patel D, Das K, Arnold E | 2013 | Crystal structure of HIV-1 reverse transcriptase (RT) in complex with Rilpivirine (TMC278, Edurant), a non-nucleoside rt-inhibiting drug | www.rcsb.org/structure/4G1Q | Publicly available at the RCSB Protein Data Bank (accession no. 4G1Q). |
| Lansdon EB | 2010 | HIV-1 Reverse Transcriptase in Complex with TMC125 | www.rcsb.org/structure/3MEC | Publicly available at the RCSB Protein Data Bank (accession no. 3MEC). |

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
