## [Decision Letter]

Thank you for submitting your article "Structural basis for potent and broad inhibition of HIV-1 RT by thiophene[3,2-*d*]pyrimidine non-nucleoside inhibitors" for consideration by *eLife*. Your article has been reviewed by three peer reviewers, and the evaluation has been overseen by a Reviewing Editor and Arup Chakraborty as the Senior Editor. The following individual involved in review of your submission has agreed to reveal his identity: Aneel Aggarwal (Reviewer #2).

The reviewers have discussed the reviews with one another and the Reviewing Editor has drafted this decision to help you prepare a revised submission.

Summary:

In this work, high-resolution crystal structures of wt and mutant HIV reverse transcriptase (RT) in complex with two newly discovered inhibitors (K-5a2 and 25a) are presented. Both bind to a pocket near to the polymerase active site. The authors have determined high-resolution crystal structures of K-5a2 and 25a with both wt and mutant RT. The authors suggest that K-5a2 and 25a were able to retain antiviral activity against some RT mutants that they claim cause a loss of RPV potency (RPV is the clinically relevant compound). However, the data they present for RPV is in apparent conflict with data in the literature. From the structures, the authors have proposed models to explain the differential susceptibility of K-5a2 and 25a and RPV to certain mutations. If the authors can resolve the differences in their antiviral data, and the antiviral data in the literature, this work should be valuable in developing HIV inhibitors with improved drug-resistance profiles.

Essential revisions:

1) The data provided in Table 1, describing the behavior of the mutants in the presence of RPV, does not appear to match the data in the literature. For example, in Table 1 the reduction in the susceptibility of the Y188L and V106A/F227L mutants to RPV is reported to be about 80-fold. However, data from at least three labs (Azijn et al., 2010; Johnson et al., 2012, Retrovirology, 9, 99; and Giacobbi and Sluis-Cremer, 2017) shows only about a 2-4 decrease in the RPV susceptibility of the Y188L mutant (because there are differences in the assays, it is more appropriate to compare fold changes in susceptibility than it is to compare absolute EC_50_ values). In addition, although the differences are less, the effect of the K013N/Y181C mutation is greater in Table 1 than in the literature (about 11 fold vs. 4-5 fold). These apparent differences may affect the interpretation of the structural data in terms of how the interactions of 25a and RPV with RT (WT and mutant) make 25a superior to RPV. Please reconcile these differences.

2) Assuming that the authors can resolve the discrepancies in their data (Table 1) and the data in the literature (see comment 1) an interesting question is why K-5a2 and 25a can apparently overcome the Y188L mutation, which they claim makes RT resistant to other NNRTIs (although data from several labs shows that Y188L does not cause a significant loss of potency for RPV). Is it possible to speculate as to why the K-5a2 and a25 can overcome the drug mutation Y188L?

3) To show that the new compound, 25a, is really better than RPV, in terms its ability to cope with resistance mutations, one should choose mutations against which RPV loses considerable potency. Looking at the data in the three papers cited in point 1, it would have been better to have tested either Y181V or I, rather than Y181C; the literature suggests that RPV loses considerably more potency against the V and I derivatives than against Y181C. Similarly, the double mutant K103N/Y181I would have been a much better choice than K103N/Y181C. Based on data in Smith et al. (2016) as well as the three abovementioned papers, it would seem reasonable to test the double mutant K101P/V179I, or some closely related variant.

4) In order to claim that 25a is superior to RPV, an important test would be to see which mutants 25a selects, and ask if RPV is more effective against them than 25a. In the absence of those data, a simple test would be to test mutations which the structure suggests might be selective for 25a. Based on the model of Kang et al. (2017), and the structures presented in the manuscript, it would seem reasonable to make mutations in the portion of the enzyme that is contacted by 25a, but not by RPV. Obvious choices would include residues K104, V106, and P236.

---

## [Author Response]

Essential revisions:1) The data provided in Table 1, describing the behavior of the mutants in the presence of RPV, does not appear to match the data in the literature. For example, in Table 1 the reduction in the susceptibility of the Y188L and V106A/F227L mutants to RPV is reported to be about 80-fold. However, data from at least three labs (Azijn et al., 2010; Johnson et al., 2012, Retrovirology, 9, 99; and Giacobbi and Sluis-Cremer, 2017) shows only about a 2-4 decrease in the RPV susceptibility of the Y188L mutant (because there are differences in the assays, it is more appropriate to compare fold changes in susceptibility than it is to compare absolute EC_50_ values). In addition, although the differences are less, the effect of the K013N/Y181C mutation is greater in Table 1 than in the literature (about 11 fold vs. 4-5 fold). These apparent differences may affect the interpretation of the structural data in terms of how the interactions of 25a and RPV with RT (WT and mutant) make 25a superior to RPV. Please reconcile these differences.

Because the data for the inhibition of RPV against HIV-1 containing Y188L RT and V106A/F227L RT had low goodness-of-fit to the inhibition dose-response curves and wide 95% confidence intervals of their EC_50_ values, we have removed the results of the two mutants from Table 1 in the revision. Instead, we have performed the in vitro RT inhibition assay using purified Y188L RT and V106A/F227L RT. The data from the in vitro RT inhibition assay have excellent fit to the inhibition dose-response curve (R^2^ ≥ 0.99) and narrow confidence intervals (Figure 4A and 4B). Our results have shown that Y188L RT and V106A/F227L RT cause 2.2-fold and 4.0-fold change in the IC_50_ values of RPV, respectively (Table 2), which are consistent with the data in the three above mentioned references. By contrast, the two mutants cause lower level of resistance to 25a (0.70- and 1.7-fold change in the IC_50_ values). Additionally, we have added the crystal structure of 25a in complex with V106A/F227L RT in the revision. Although there are some changes in the binding mode of 25a caused by the V106A/F227L double-mutation, the buried surface area between 25a and the NNIBP of V106A/F227L RT is similar to that between 25a and WT RT (Figure 5, Table 4). The structure provides an explanation of the low impact of V106A/F227L on the potency of 25a.

With respect to the K103N/Y181C mutation, the discrepancies in the fold change of RPV’s potency between our data and the data from other literatures could arise from the differences in the cellular assay systems, in which the impact of mutations on drug potency can be complicated by other viral and host factors. To provide further evidence, we have tested the RT-inhibitory potency of 25a and RPV against K103N/Y181C RT using in vitro RT inhibition assay and found that 25a (7.2-fold change in IC_50_) is more resilient to the double-mutation than RPV (15-fold change in IC_50_) (Figure 4A, 4B and Table 2). These results are in good agreement with the results from our T-cell based antiviral assay. Moreover, since the potency of 25a and RPV was tested in the same assay system at the same time, the data should support our conclusion that 25a is superior to RPV in inhibiting Y188L RT, V106A/F227L RT and K103N/Y181C RT.

We have incorporated the new data in Figure 4A and 4B, Figure 5D, Table 2, Table 4 and Results subsection “Inhibition of HIV-1 RT by piperidine-substituted thiophene[3,2-*d*]pyrimidine NNRTIs” in the revision.

2) Assuming that the authors can resolve the discrepancies in their data (Table 1) and the data in the literature (see comment 1) an interesting question is why K-5a2 and 25a can apparently overcome the Y188L mutation, which they claim makes RT resistant to other NNRTIs (although data from several labs shows that Y188L does not cause a significant loss of potency for RPV). Is it possible to speculate as to why the K-5a2 and a25 can overcome the drug mutation Y188L?

We have determined the crystal structure of K-5a2 in complex with Y188L RT and compared the binding modes of K-5a2 in its complexes with WT RT and Y188L RT in our previous submission. In the revision, we have analyzed the binding of K-5a2 to Y188L RT in comparison with its binding to WT RT in further detail:

“In the structure of Y188L RT/K-5a2 complex, the cyano-dimethylphenyl group in K-5a2 is diverted away from Leu188 to avoid steric clashes, leading to declines in the buried areas between the inhibitor and Leu188 and Phe227. However, this mutation-caused damage is alleviated by its enhanced interactions with Lys103, Val106 and Pro236 (Figure 5—figure supplement 1C, Table 4).”

Based on these observations, we have speculated the reasons why the Y188L mutation may have lesser effects on K-5a2 and 25a than on RPV:

“The improved resistance profiles of 25a over RPV against other RT mutants, especially Y181I RT, Y188L RT and K103N/Y181C RT, are likely due to the bulkier substituents in its right wing and more extensive hydrogen bond interactions with NNIBP residues. […] Moreover, the higher degree of strategic flexibility of 25a (because of more rotatable bonds) can make it more effective to preserve or even enhance its interactions with other NNIBP residues when Y181I and Y188L mutations displace the left wing structure of 25a.”

3) To show that the new compound, 25a, is really better than RPV, in terms its ability to cope with resistance mutations, one should choose mutations against which RPV loses considerable potency. Looking at the data in the three papers cited in point 1, it would have been better to have tested either Y181V or I, rather than Y181C; the literature suggests that RPV loses considerably more potency against the V and I derivatives than against Y181C. Similarly, the double mutant K103N/Y181I would have been a much better choice than K103N/Y181C. Based on data in Smith et al. (2016), as well as the three abovementioned papers, it would seem reasonable to test the double mutant K101P/V179I, or some closely related variant.

We thank the reviewers for this constructive suggestion. In the revision, we have tested the inhibitory potency of 25a and RPV against purified K101P RT, Y181I RT and K103N/Y181I RT. While all three mutations substantially reduce susceptibility to RPV (20-fold, 90-fold and 1805-fold change in the IC_50_ values), they cause considerably less resistance to 25a (1.3-fold, 8.8-fold and 96-fold change, respectively). The results suggest that 25a is much more effective than RPV in inhibiting K101P RT, Y181I RT and K103N/Y181I RT. These new results have been added to Figure 4C, 4D, Table 2 and the Results section.

To understand why 25a can overcome these mutations, we have determined the crystal structures of 25a in complex with K101P RT and Y181I RT. The attempt to obtain the crystal structure of 25a in complex with K103N/Y181I RT was unsuccessful, likely due to its suboptimal inhibitory potency towards K103N/Y181I RT, although it has displayed marked improvement over RPV against this specific mutant. The new structural data have been incorporated in Figure 2—figure supplement 1, Figure 5, Figure 6—figure supplement 1, Table 3 and Table 4. The related structural analyses have been added in the Results section:

“With respect to K101P RT, the deprotonation of Pro101 main-chain nitrogen attenuates its water-mediated hydrogen bond with the pyrimidine nitrogen in 25a. […] This unfavorable change in the NNIBP is mitigated by the enlarged contact areas between 25a and the more closely placed side-chains of Lys101 and Tyr183 (Figure 5F).”

We have also added the following discussion to the subsection “Comparison of the binding modes for K-5a2, 25a and DAPY NNRTIs”:

“The K101P mutation considerably reduces susceptibility to RPV. Superposition of the WT RT/RPV and K101P RT/25a complex structures reveals that the mutation would remove the hydrophobic interaction between the RPV pyrimidine ring and the long aliphatic side chain of Lys101, and possibly introduces steric clashes between the polar groups in the pyrimidine ring and the non-polar side chain of Pro101. More importantly, Lys to Pro substitution will abrogate the critical hydrogen bond between the pyrimidine nitrogen of RPV and the backbone nitrogen in RT due to deprotonated form of Pro101 main-chain nitrogen and hence its inability to act as a hydrogen bond donor (Figure 6—figure supplement 1C).”

4) In order to claim that 25a is superior to RPV, an important test would be to see which mutants 25a selects, and ask if RPV is more effective against them than 25a. In that absence of those data, a simple test would be to test mutations which the structure suggests might be selective for 25a. Based on the model of Kang et al. (2017), and the structures presented in the manuscript, it would seem reasonable to make mutations in the portion of the enzyme that is contacted by 25a, but not by RPV. Obvious choices would include residues K104, V106, and P236.

This is a good point raised by the reviewers. Based on our structure, the longer right wing of 25a interacts with Pro225 and Pro236, both of which are not utilized by RPV in its binding to RT. Therefore, in the revision, we have assessed the inhibitory potency of 25a against RT containing P225H or P236L substitutions, two clinically identified mutations shown no significant reduction in susceptibility to RPV (Basson et al., 2015). Like that of RPV, the potency of 25a was not negatively affected by either P225H or P236L mutations (0.58- and 0.60-fold change in IC_50_, respectively) (Figure 4E, 4F and Table 2), indicating that 25a has a relatively high genetic barrier to the development of novel drug-resistant mutations. The new results have been added to Figure 4E, 4F, Table 2 and the Results section.

We have also explained why mutations on K104 and V106 are unlikely to cause significant reductions in susceptibility to 25a in the Discussion section:

“Additional RT-25a interactions that are not present in the RT/RPV complex include the hydrogen bonds between the sulfonamide group of 25a and Lys104 and Val106. […] Even if the side-chain substitutions deviate the main chains, a minor change in the torsion angle τ1, which was shown to span a wide range without significant energetic penalty (Table 5), would readily place the sulfonamide group of 25a in an optimal position for hydrogen bonds formation.”